# PeerJ

# Associating disease-related genetic variants in intergenic regions to the genes they impact

Geoff Macintyre[1,6,7], Antonio Jimeno Yepes[1,7], Cheng Soon Ong[2,3,4] and Karin Verspoor[1,5]

[1] Department of Computing and Information Systems, The University of Melbourne, VIC, Australia
[2] Department of Electrical and Electronic Engineering, The University of Melbourne, VIC, Australia
[3] Machine Learning Group, NICTA Canberra Research Laboratory, Australia
[4] Research School of Computer Science, Australian National University, Australia
[5] Health and Biomedical Informatics Centre, The University of Melbourne, VIC, Australia
[6] Centre for Neural Engineering, The University of Melbourne, VIC, Australia
[7] These authors contributed equally to this work.

## ABSTRACT

We present a method to assist in interpretation of the functional impact of intergenic disease-associated SNPs that is not limited to search strategies proximal to the SNP. The method builds on two sources of external knowledge: the growing understanding of three-dimensional spatial relationships in the genome, and the substantial repository of information about relationships among genetic variants, genes, and diseases captured in the published biomedical literature. We integrate chromatin conformation capture data (HiC) with literature support to rank putative target genes of intergenic disease-associated SNPs. We demonstrate that this hybrid method outperforms a genomic distance baseline on a small test set of expression quantitative trait loci, as well as either method individually. In addition, we show the potential for this method to uncover relationships between intergenic SNPs and target genes across chromosomes. With more extensive chromatin conformation capture data becoming readily available, this method provides a way forward towards functional interpretation of SNPs in the context of the three dimensional structure of the genome in the nucleus.

Corresponding author
Karin Verspoor,
karin.verspoor@unimelb.edu.au

## INTRODUCTION

Detection and characterisation of disease-associated genetic variation is a major emphasis of current scientific inquiry. Recent technological advances have enabled investigation of increasingly complex genetic architectures and their relationship to disease. Genome-wide association studies (GWAS) are using large sample cohorts to associate increasing numbers of single-nucleotide polymorphisms (SNPs) with disease phenotypes (*Solovieff et al., 2013*), and whole-genome sequencing efforts such as The Cancer Genome Atlas (*The Cancer Genome Atlas Research Network, 2011*) are uncovering large numbers of putative cancer-causing single nucleotide variations. However, the functional interpretation

of these variations has focused mainly on those residing in protein-coding regions of the genome (*Ward & Kellis, 2012b*), despite the fact that the majority lie in non-coding regions (*Pastinen, 2010*; *The 1000 Genomes Project Consortium, 2010*). This is due to some of the difficulties in functional interpretation of intergenic SNPs. Distinguishing driver mutations from passenger in intergenic regions is difficult and intergenic SNPs may affect function in subtle ways such as disrupting regulatory elements (*Schaub et al., 2012*). Experimental approaches for validation of the functional impact of intergenic variation are involved (*Paul, Soranzo & Beck, 2014*), thus new computational approaches are emerging in an attempt to reduce the burden of functional interpretation of intergenic variation (*Ward & Kellis, 2012a*; *Macintyre et al., 2010*; *Li et al., 2013*).

Functional interpretation of an intergenic variant requires that the gene impacted by the variant be identified—the target gene (*Anonymous, 2012*). A common mode of impact is via disruption of a regulatory element (enhancer) controlling the target gene (*Paul, Soranzo & Beck, 2014*). However, as regulatory elements can be located far away from the genes they regulate (*Lieberman-Aiden et al., 2009*; *Fullwood et al., 2009*), associating variant and target gene can be problematic. Considering genes flanking the variant in linear genomic space will, in many cases, miss the target gene. However, considering larger search spaces will often result in too many putative targets. Therefore additional information is required to associate the variation with its target gene.

Chromatin marks (*Ernst & Kellis, 2010*), DNA conservation (*Spivakov et al., 2012*), chromatin interaction data (*Lieberman-Aiden et al., 2009*), expression quantitative trait loci (eQTL) (*Gilad, Rifkin & Pritchard, 2009*), changes in transcription factor binding affinities (*Macintyre et al., 2010*), and combinations of these (*Li et al., 2013*; *Duggal, Wang & Kingsford, 2014*), have been shown to succesfully reduce the target gene search space. In many cases, however, the underlying data is such that there is inadequate statistical power to achieve completely unbiased search. For example, eQTLs are based on a statistical association test between genetic variants and gene expression levels and hence their determination depends on having sufficient sample sizes (typically tens or hundreds of individuals with both sequence and gene expression data). Therefore eQTL studies are usually carried out considering only genes nearby SNPs and as such this data is biased towards SNP-proximal gene associations (*Nica & Dermitzakis, 2013*).

Careful selection of the types of data used for search space reduction is required in order to achieve an unbiased search strategy. Recent experimental advances in mapping the three dimensional structure of the genome in the nucleus have led to the HiC method (*Lieberman-Aiden et al., 2009*). These data are ideal for limiting the search space for candidate target genes of a variant in an unbiased fashion. For any given genomic region containing a variant, it is possible to determine what other regions in the genome are in close proximity within the nucleus. It can be assumed that these regions are more likely to contain a gene which is a good candidate for being a direct target of an intergenic variant, and indeed recent work has demonstrated that chromatin structure is useful for characterising eQTLs (*Duggal, Wang & Kingsford, 2014*). However, the current resolution of the genomic interaction data from these technologies is such that the list of candidate

genes obtained from regions that interact with the region containing the variant is too large. Therefore, more information is required to prioritise this candidate list. The function of candidate target genes of a variant may in many cases have been explored within the biomedical literature in the context of a disease. Consequently, it should be possible to mine the literature for this information and prioritise candidate genes accordingly.

In this work, we explore strategies for ranking the association of a DNA variant in an intergenic region to a gene, without the need for SNP-related gene expression data (eQTLs), but using evidence that is more informative than simple sequence distance. We consider two primary approaches that take advantage of general background knowledge to establish putative associations between intergenic variants and genes. One source of this knowledge is the published biomedical literature, which is mined to establish SNP-gene relationships, considered in the context of specific diseases. This can be viewed as capturing the broader, indirect associations among SNPs and genes. Another source is the 3D "interactome" data that is indicative of physical interactions at the genomic level.

We compare these strategies, both individually and in combination, to a sequence distance baseline, with respect to a set of eQTLs. Our results are suggestive of the promise of the approach, with the proposed hybrid ranking scheme outperforming the distance baseline , and an exploratory analysis identifying a putative functional mechanism for a gene ranked as a high-ranked candidate target for a given SNP in the data set.

## METHODS

When attempting to interpret the functional impact of an intergenic SNP, a number of approaches are typically employed. As a first step, the likely target gene of the SNP is determined. Once this has been achieved, the relationship of the SNP relative to the putative target gene will dictate which experimental or computational procedures will be used to verify the relationship. For intergenic SNPs it is generally considered that they will have some sort of regulatory effect. Candidate target gene search typically involves looking at the closest gene to the SNP in linear genomic distance. Given that genetic units are inherited as blocks, it is not a bad assumption to make that the nearest gene is the most likely candidate. However, given that chromatin looping plays a significant role in gene regulation (*Fullwood et al., 2009*), the chromatin architecture may be such that the nearest gene is not the correct target gene. As soon as many genes are considered putative targets, approaches are required to rank these genes for subsequent followup. Below, we explain four ranking approaches considered in this paper.

### Genome sequence distance (2D) ranking

Given the fundamental assumption that intergenic SNPs impact the closest genes to them, as measured by sequence distance, we have employed a baseline ranking method based on genomic sequence distance. In this method, the association between an intergenic SNP and a gene is the number of intervening base pairs separating them. The ranking is determined by the distance, from the smallest distance to the largest. Genes on different chromosomes are given the bottom rank as no 2D distance can be calculated. Gene annotations from

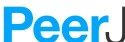

[1] dbSNP: https://www.ncbi.nlm.nih.gov/SNP/

build hg18 for the human genome were used to compute distances from the SNPs of interest from dbSNP 132.[1]

## Spatial (3D) ranking

The HiC method proposed by *Lieberman-Aiden et al. (2009)* allows unbiased identification of chromatin interactions across an entire genome, providing a physical basis for determining interactions among genomic regions. In brief, chromatin is cross-linked with formaldehyde then restriction enzyme digested, and the remaining fragments are ligated so that linked regions form the same ligation product. These products are then subjected to deep sequencing. The method produces a probability contact map for genomic regions, at megabase resolution. The strength of probability among the region of a given intergenic SNP of interest and the location of a gene in this 3D map is used to establish the spatial association between the intergenic SNP and any gene it is in contact with (subject to a threshold). We used HiC profiling of lymphoblast cells (*Lieberman-Aiden et al., 2009*) subjected to the normalisation method of *Yaffe & Tanay (2011)* which computes log (base2) ratios of observed and expected read counts for each 1 MB window across the human genome (build hg18). For any given SNP, the log ratio of all 1 MB regions across the genome interacting with the 1 MB region containing the SNP was assigned to genes falling within these regions. For the given SNP, all genes were then ranked according to their likelihood of interacting with the SNP based on HiC profiling. In this study, we limited our analysis to HiC data for the HindIII restriction enzyme. This source of data is referred to as $rank_{spatial}$.

## Literature-based ranking

We develop two literature-based ranking strategies, that capture complementary information.

The first approach (described in "Gene-disease association scores") strictly focuses on establishing disease-specific rankings of genes. The rankings reflect the strength of association between a disease and a set of genes, based on how frequently a disease and a gene are mentioned together in the literature (with respect to background expectation for their co-mention). The particular SNPs of interest do not play a specific role in this analysis, other than to limit the set of diseases considered; indeed the ranking of genes will be the same for every SNP, with respect to a given disease.

The second approach (described in "SNP-gene co-occurrence score ranking") aims to identify direct support for the relationship between an SNP and a gene, by searching the literature specifically for abstracts in which an SNP and a gene are mentioned together. This approach is clearly tied to the availability of abstract-level details about an SNP existing in PubMed; this data is very sparse, in part due to the fact that specific details about genetic variants are only rarely available in published abstracts (*Jimeno Yepes & Verspoor, 2014a*). Unfortunately, the access to the full content of the article is limited, with only around 600k articles available from PubMed Central (a small proportion of the over 23M citations available from PubMed). However, where it is available it can provide important evidence of an association.

Both methods are applied to a corpus of documents which is processed to detect mentions of genes and dbSNP SNPs, and to identify the diseases that specific publications are related to. More details of these steps are provided below.

Several prior efforts have addressed the interpretation of SNPs via the biomedical literature. The GRAIL method (*Raychaudhuri et al., 2009*) focuses on prioritising genes in disease-implicated regions on the basis of similarities of genes as determined by the similarity of the texts associated to the genes. The method has the effect of aggregating all genes within a given disease region, though the analysis usually results in a single candidate gene in each region being recognised as most significant. Determination of the genes in a region of an SNP in this method is based on LD (linkage disequilibrium) characteristics, and is effectively based on genomic distance. Similarly, the method of *Johansson et al. (2012)* collects the genes within a 50k base pair window around an SNP, and then determines probabilities of association of those genes to a disease or phenotype based on key terms in the literature associated to the genes. These methods are both successful in supporting SNP interpretation, and demonstrate the utility of the literature in supporting functional analysis. However, they assume physical proximity of SNPs to genes as the overriding basis for associating SNPs to genes; we will show that by considering a broader set of genes we can identify relationships they may miss.

### Literature sources

Resources from the US National Library of Medicine (NLM) like PubMed and PubMed Central offer access to the biomedical literature. PubMed currently indexes over 23 million biomedical publications, and provides publication meta-data (title, authors, journal, publication date) as well as associated index concepts from the Medical Subject Headings (MeSH) controlled vocabulary. Among these concepts is the general concept "cancer". We used this term to identify cancer-related literature in PubMed. PubMed expanded this query automatically to include MEDLINE citations indexed with the MeSH heading "Neoplasms" and any descendent term in the MeSH hierarchy. We retrieved 2.7 million cancer-related citations from PubMed in December 2012 using this term, and built a local repository of the abstract text of each of these citations. We refer to this as the *cancer corpus*. We have restricted our analysis in this paper to cancer related diseases for two reasons. Firstly, it allows us to limit our predictions of SNP-gene interaction to a managable number for evaluation of performance of the method. Secondly, the cancer literature is extensive and mature, and includes substantial SNP association studies providing an adequate number of instances in which to assess performance of our approach.

### Intergenic SNP mention detection

The database dbSNP (*Sherry et al., 2001*) provides large catalog of SNPs, publicly available from the NCBI. It provides a mechanism for specifically identifying mutations that occur in intergenic regions as explained on their website.[2] We applied that procedure to retrieve all human intergenic SNPs in dbSNP Build 139, available online as of October 31, 2013. From these, we sub-select only those that are mentioned in the literature corpus that we have compiled.

[2] http://www.ncbi.nlm.nih.gov/books/NBK44466//#Search.how_do_i_obtain_snps_located_in_h

Since our study is limited to intergenic SNPs identified through dbSNP, we relied exclusively on mentions of the dbSNP identifiers, also known as the rsIDs (because each is in the form "rs00000"). The EMU tool (*Doughty et al., 2011*) was used to detect mention of rsIDs in the abstracts. 10,143 rsID mentions were detected in the set of 2.7 million abstracts, containing 5,337 unique rsIDs. Of these, 2,409 mentions were classified as intergenic based on dbSNP, representing 1,181 unique rsIDs.

### Gene mention detection and normalisation

The literature is used in our work to establish background statistics of gene-disease association. The starting point for this is detection of mentions of specific genes in the literature, and normalisation of those mentions to NCBI Gene database identifiers. We used a dictionary-based approach to annotate the corpus, as such methods have been shown to be effective for gene normalisation (*Jimeno-Yepes et al., 2013*; *MacKinlay & Verspoor, 2013*), especially for the context of interpreting human SNPs, since human genes are less ambiguous than gene names in other species.

We used a large dictionary of gene names based on a dictionary built from the human subset of the NCBI Gene database. We followed the procedure in *Jimeno-Yepes et al. (2013)*, removing duplicates and filtering out certain misleading or ambiguous gene names, such as those ending with "disease", "syndrome", or "susceptibility", and removed terms from a standard stopword list.[3] The entries in the dictionary are linked to their originating NCBI Gene identifier. If an entry is common to multiple NCBI Gene records, all relevant identifiers are associated with the dictionary entry.

We used the ConceptMapper tool (*Tanenblatt, Coden & Sominsky, 2010*; *Funk et al., 2014*)[4] as the dictionary tagger tool, reusing the configuration prepared for the BioCreative 2013 CTD track (*MacKinlay & Verspoor, 2013*), which does not make case distinction, tokens have to be matched in the same order, only the longest match is considered and tokens that form part of a multiple-token name must be adjacent to each other. The output of ConceptMapper provides a list of extracted genes with their position in text and their normalization to an NCBI identifier.

ConceptMapper with the NCBI derived dictionary was used to annotate the MEDLINE cancer subset. For each gene mention identified in text, we have the PMID of the citation, the start and end position of the mention, the term identified in text and its NCBI identifier. From this annotation, we extract the genes identified in each citation. This list is used to derive gene-disease associations as shown below.

### SNP-disease relationships

We utilise the literature to establish an association between an SNP and a disease. We first apply the intergenic SNP detection described above. For each citation in our corpus that mentions at least one intergenic SNP, we then look for any cancer-related MeSH term in the meta-data for that citation. MeSH indexing of MEDLINE summarizes the most relevant MeSH concepts with respect to the content of the article and it is available from the MEDLINE baseline distribution. Cancer-related MeSH terms are considered the ones under the *Neoplasms* MeSH heading with tree number starting with *C4*. This provides us

[3] Stopword list from the SMART system: ftp://ftp.cs.cornell.edu/pub/smart.

[4] http://uima.apache.org/d/uima-addons-current/ConceptMapper/ConceptMapperAnnotatorUserGuide.html

with SNP-disease relations. The diseases from these relations are used as a disease filter; we estimate gene-disease associations for these diseases as explained below.

### Gene-disease association scores

To establish the strength of a gene-disease association, we must first count the number of times each gene is mentioned in an abstract related to a given disease throughout our corpus. We apply the gene normalisation step to find all abstracts that mention a given gene. For each abstract in our corpus that mentions at least one gene, we then look for any cancer-related MeSH term in the meta-data for that abstract following the same procedure as for SNP-disease relationships. For each such disease MeSH term, we record an association between the genes mentioned in the abstract, and that disease. In this manner, we establish counts for each {gene, disease} pair in the corpus.

The gene-disease association score is estimated based on the $t$-test, where the test looks at the difference between the observed $\bar{x}$ and the expected mean $\mu$. We have preferred this to other alternatives like mutual information due to the problems that they have with respect to low frequency events (*Manning & Schütze, 1999*).

In the test, the expected mean is the probability of the gene and the disease not being associated, i.e., being independent $P(Gene)P(Disease)$, and the observed $P(Gene, Disease)$ is the probability of the gene and the disease together. The variance is based on the observed distribution considered as a Bernoulli distribution. $N$ is the number of citations in the *cancer* corpus.

$$t = \frac{\bar{x} - \mu}{\sqrt{\frac{s^2}{N}}}. \tag{1}$$

The literature-based gene-disease association score generates a ranking that will be the same regardless of which SNP is considered since that method does not consider the SNP directly. However, we utilise the established SNP-disease relationships to limit the diseases considered relevant to the SNP. This in turn restricts the set of genes which are considered to be potentially relevant to the SNP. Thus, we produce SNP-gene associations that are based on the SNP's association to a disease, and intersect the diseases common to the SNP-disease and disease-gene sets to produce a set of SNP-disease-gene associations. Each SNP-disease-gene association is assigned the score of the underlying disease-gene association. This score is used to compute $rank_{literature}$.

### SNP-gene co-occurrence score ranking

To consider the possibility of direct mentions of intergenic SNP-gene interactions in the literature, we counted the number of abstracts in which a given intergenic SNP from our data set and a gene are mentioned together with the text of an abstract (an SNP-gene "co-mention"). Although we did not attempt to establish the specific nature of the relationship between the SNP and the gene as mentioned in the article, prior work has found that simple co-occurrence of such concepts can be useful to establish a biological relationship (*Gabow et al., 2008*; *Verspoor et al., 2013*; *Sokolov et al., 2013*).

**Table 1** Scores vs ranks: for a list of length *n* the fractional rank is given by the position *i* of the object divided by $n+1$.

| Distance | 43k | 2k | 500k | 8k | 1292k |
|---|---|---|---|---|---|
| Position (*i*) | 3 | 1 | 4 | 2 | 5 |
| Rank ($\frac{i}{n+1}$) | $\frac{3}{6} = 0.5$ | $\frac{1}{6} = 0.167$ | $\frac{4}{6} = 0.667$ | $\frac{2}{6} = 0.333$ | $\frac{5}{6} = 0.833$ |

There are very few of these intergenic SNP-gene co-mentions within the abstracts of our literature dataset. There is an average of 5.88 unique genes co-mentioned with each of the 1,181 intergenic SNPs in the data set, with a standard deviation of 7.36 across the set of SNPs. This value is consistent with the result for all the SNPs in the full dbSNP set (average 5.77 and standard deviation 7.33).

We generated a ranking over the genes co-mentioned with each SNP based strictly on the occurrence count of the co-mentions. However, we found this too sparse for direct use (see "Evaluation of performance of hybrid approach against baseline").

## Hybrid Spatial-Literature association ranking

Recognising that genomic distance, the spatial association method and the literature-based gene-disease association method provide complementary information—the physical distance methods capturing mostly direct relationships among SNPs and genes and the literature method capturing mostly indirect relationships—we experimented with a method combining them.

The hybrid method takes the ranked list of genomic distances, the ranked list of SNP-gene associations from the spatial method and the ranked list of SNP-(disease)-gene associations based on the gene-disease association method, and computes a hybrid rank for each SNP-(disease)-gene association as the geometric mean of the ranks from the three source methods (*Bedő & Ong, 2014*). This has the effect of maintaining a high overall rank for an association if it is ranked high in one of the three methods, even when the rank is very low in the other two.

Since ranks are invariant to monotone transforms of the corresponding scores, some care has to be taken when aggregating them. In particular, a normalisation procedure has to be applied to the numerical representation to account for the fact that we have lists which differ in length. In this paper, we consider fractional ranks in the unit interval, that is each object's rank is its position in the sorted list normalised by the length of the list. The simple example in Table 1 illustrates this concept for genomic distance. The corresponding operations for the spatial method and the literature method are the same.

After ranking, the different sources of information are comparable, and can be directly multiplied. Therefore, the hybrid rank of $rank_{gen\_dist}$, $rank_{spatial}$ and $rank_{literature}$ is calculated as

$$rank_{hybrid} = \text{rank}(rank_{gen\_dist} * rank_{spatial} * rank_{literature}), \qquad (2)$$

where the operation rank($\cdot$) just performs the normalisation described in Table 1. This hybrid rank is used in the validation procedure in the experiments. For the discovery experiments which are across different chromosomes, genomic distance is meaningless, and therefore is dropped from the hybrid ranking,

$$rank_{hybrid} = \text{rank}(rank_{spatial} * rank_{literature}). \tag{3}$$

## Evaluation

Evaluation of any predictive method is difficult in the absence of labeled, gold standard data. Since functional association of intergenic SNPs to genes has been studied substantially less than the impact of mutations within genes or proteins, there is limited data available to validate our methods. Therefore a gold standard proxy must be used. The most suitable data for evaluation of our methods in this case comes from eQTL studies. These data provide links between SNPs and putative target genes based on statistical assocation of alleles and gene expression. It has been previously shown that these associations have a strong link with 3D genome structure linking both genes and SNPs nearby in linear and 3D space (*Duggal, Wang & Kingsford, 2014*). From the eQTL data, we have for each SNP a target gene implicated as the gene affected by that SNP. We take this as our gold standard proxy—the gene that each method should rank high in the set of potentially associated genes.

We consider two different experiments to demonstrate the utility of an unbiased search strategy: a validation experiment to compare the performance of our proposed method with a genomic distance baseline, and a discovery experiment where we focus on interchromosomal associations. For the validation experiment, we had to limit the set of genes considered to those residing on the same chromosome as the SNP since the genomic distance baseline could not be computed between chromosomes. In addition, we focussed the comparison on long-range enhancer-like associations by excluding genes within 500 Kbp of the SNP under consideration. For the discovery experiment, we considered only target genes from different chromosomes as the SNP under study to determine if cross-chromosomal interactions could be detected.

### eQTL data sources

For evaluaton of our approach we extracted eQTL data from the GTEx (Genotype-Tissue Expression) eQTL Browser (*Lonsdale et al., 2013*) across all eQTL studies in the database. We obtained all eQTLs which matched SNPs in our disease-SNP dataset mined from the literature. In each case, the associated gene was recorded along with the reported *p*-value for the association between the SNP and gene. This resulted in a set of 26 unique SNPs for which there was a significant ($p < 0.01$) association with the expression of a gene. After filtering of these eQTL SNPs, 17 significant associations were considered positive labels in our ranking assessment of the SNP target gene methods.

For confirmation of discovery of novel SNP target gene pairs, we queried the HapMap3 (*Stranger et al., 2012*) eQTL study data via the GeneVar database.[5] If a significant

[5] http://www.sanger.ac.uk/resources/software/genevar

association between disease SNP and putative target gene was found, this was considered evidence for a putative functional relationship between the SNP and gene.

### Comparison of rankings

For each SNP, the eQTL data provides a gene that is strongly associated to that SNP. Our analysis compares the rank of that target gene in the ranking produced by each method.

We consider the genomic distance-based ranking as the baseline ranking to which we compare the alternative ranking methods. This reflects the fact that genomic distance is the primary method used in current approaches to establish SNP-gene associations, as well as the importance of DNA proximity at a molecular level. In addition, we compute the ranking based on the HiC likelihood and the literature based sources. For each SNP-disease pair, we ordered all genes based on each of the three sources of information above, and identified the ranks at which there is eQTL evidence for association. The spatial and literature rankings are then aggregated using the geometric mean (Eq. (2)).

## RESULTS

### Evaluation of performance of hybrid approach against baseline

Using eQTL associations between the disease SNPs and genes as a gold standard (per "Evaluation") we assessed the performance of our hybrid ranking methods against a baseline ranking based on 2D (linear) genomic distance. As mentioned before, we focus on long range associations ($>500$ Kbp) since these are the cases of interest. Furthermore the 1 MBps resolution of the HiC data means that the score derived from HiC could be on average up to 500 Kbp distant from the SNP of interest. For fair comparison with genomic distance we limited the hybrid rank approach to genes on the same chromosome as the SNP. This resulted in a total of 81,569 associations, of which only 18 had eQTL evidence. Two SNPs (rs2029166 and rs7296239) within 2 Kbp from each other in AAAS gene had identical spatial and literature scores hence we considered them as one single instance, leaving a total of 17 eQTL for assessing performance.

The details of the rankings are shown in Table 2. Note that the rankings are performed for each SNP-disease pair, hence a particular gene may be ranked the same for different SNP-disease combinations where the ranking relies on same HiC experiment (for example rs344781 in gene PSG11). This shows an additional advantage of the literature-based approach which allows prioritisation of the putative associations using disease specific information. From Table 1, it can be seen that spatial ranking (HiC ranking) or literature ranking alone do not provide results comparable to ranking based on linear genomic distance. However, a combination of both methods (hybrid ranking) outperforms linear ranking. Figure 1 highlights this by plotting, for each of the eQTL SNPs, the rank of the target gene based on genomic distance ($rank_{gen\_dist}$) against the rank of that target gene based on $rank_{hybrid}$. Three of the 17 eQTLs had lower ranks, and only the top 14 eQTLs are shown. Better performance corresponds to lower rank, i.e., if the rank according to the hybrid method is lower than the rank of the genomic distance, then the hybrid method has outperformed genomic distance-based ranking. This is the case for twelve eQTL SNPs, while $rank_{gen\_dist}$ outperforms $rank_{hybrid}$ in five cases. Hence the hybrid method has an

**Table 2 eQTL data used for validation.** The eQTL column shows the *p*-value of the eQTL data. GD means genomic distance, and Lit. refers to the score resulting from literature. rD, rHiC and rL refer to the ranks of genomic distance, HiC and literature respectively, out of Total number of genes which are more than 500 Kbp from the SNP under consideration. rHy is the geometric mean $rank_{hybrid}$. A nan value in Lit. means that no evidence was found in the literature.

| PubMedID | SNP | Disease | chr | Location | Gene | GeneID | GeneWindow | eQTL | GD | HiC | Lit. | rD | rHiC | rL | rHy | Total |
|---|---|---|---|---|---|---|---|---|---|---|---|---|---|---|---|---|
| 21995493 | rs652625 | Carcinoma, NSC Lung | chr1 | 12147937 | MTOR | 2475 | chr1:11M-12M | 3.94E-05 | 902741 | 6.84 | 1.17 | 0.0091 | 0.0046 | 0.0282 | 0.0012 | 2408 |
| 20937265 | rs344781 | Carcinoma, NSC Lung | chr19 | 48866627 | PSG11 | 5680 | chr19:48M-49M | 1.96E-05 | 644155 | 7.16 | nan | 0.0031 | 0.0006 | 0.2842 | 0.0013 | 1594 |
| 21761413 | rs344781 | Endometrial Neoplasms | chr19 | 48866627 | PSG11 | 5680 | chr19:48M-49M | 1.96E-05 | 644155 | 7.16 | nan | 0.0031 | 0.0006 | 0.2748 | 0.0013 | 1594 |
| 20937265 | rs344781 | Lung Neoplasms | chr19 | 48866627 | PSG11 | 5680 | chr19:48M-49M | 1.96E-05 | 644155 | 7.16 | nan | 0.0031 | 0.0006 | 0.2992 | 0.0013 | 1594 |
| 19760037 | rs7187167 | Breast Neoplasms | chr16 | 1289209 | WDR24 | 84219 | chr16:0-1M | 2.73E-05 | 608807 | 7.21 | nan | 0.0156 | 0.0021 | 0.7419 | 0.0042 | 961 |
| 21995493 | rs652625 | Lung Neoplasms | chr1 | 12147937 | MTOR | 2475 | chr1:11M-12M | 3.94E-05 | 902741 | 6.84 | −5.59 | 0.0091 | 0.0046 | 0.2263 | 0.0046 | 2408 |
| 23059779 | rs12983047 | Carcinoma, Squamous Cell | chr19 | 46526338 | CIC | 23152 | chr19:47M-48M | 0.00036 | 954318 | 7.15 | 1.42 | 0.0218 | 0.0230 | 0.0131 | 0.0075 | 1607 |
| 19358266 | rs7187167 | Lymphatic Metastasis | chr16 | 1289209 | WDR24 | 84219 | chr16:0-1M | 2.73E-05 | 608807 | 7.21 | nan | 0.0156 | 0.0021 | 0.7034 | 0.0083 | 961 |
| 23059779 | rs12983047 | Adenocarcinoma | chr19 | 46526338 | CIC | 23152 | chr19:47M-48M | 0.00036 | 954318 | 7.15 | 0.33 | 0.0218 | 0.0230 | 0.0492 | 0.0137 | 1607 |
| 23059779 | rs12983047 | Lung Neoplasms | chr19 | 46526338 | CIC | 23152 | chr19:47M-48M | 0.00036 | 954318 | 7.15 | −0.61 | 0.0218 | 0.0230 | 0.0834 | 0.0168 | 1607 |
| 22267197 | rs2823093 | Breast Neoplasms | chr21 | 15442702 | USP25 | 29761 | chr21:16M-17M | 0.00010 | 581512 | 6.57 | −1.75 | 0.0101 | 0.0709 | 0.1723 | 0.0169 | 296 |
| 22344756 | rs3213182 | Carcinoma, Squamous Cell | chr20 | 31726893 | ITCH | 83737 | chr20:32M-33M | 0.00021 | 687808 | 6.95 | −5.09 | 0.0142 | 0.0236 | 0.2079 | 0.0173 | 635 |
| 23059779 | rs12983047 | Carcinoma, NSC Lung | chr19 | 46526338 | CIC | 23152 | chr19:47M-48M | 0.00036 | 954318 | 7.15 | −2.16 | 0.0218 | 0.0230 | 0.1176 | 0.0243 | 1607 |
| 21427733 | rs2029166 | Breast Neoplasms | chr12 | 52876365 | AAAS | 8086 | chr12:52M-53M | 7.79E-05 | 874685 | 6.40 | −2.42 | 0.0131 | 0.0533 | 0.2238 | 0.0262 | 1144 |
| 21427733 | rs7296239 | Breast Neoplasms | chr12 | 52877970 | AAAS | 8086 | chr12:52M-53M | 0.00011 | 876290 | 6.40 | −2.42 | 0.0131 | 0.0533 | 0.2238 | 0.0262 | 1144 |
| 22344756 | rs3213182 | Head and Neck Neoplasms | chr20 | 31726893 | ITCH | 83737 | chr20:32M-33M | 0.00021 | 687808 | 6.95 | nan | 0.0142 | 0.0236 | 0.7858 | 0.0283 | 635 |
| 21733090 | rs7096206 | Carcinoma, Hepatocellular | chr10 | 54201690 | CHUK | 1147 | chr10:101M-102M | 1.43E-06 | 47777644 | 2.00 | −1.08 | 0.6705 | 0.9153 | 0.1695 | 0.5931 | 956 |
| 21733090 | rs7096206 | Liver Neoplasms | chr10 | 54201690 | CHUK | 1147 | chr10:101M-102M | 1.43E-06 | 47777644 | 2.00 | −4.22 | 0.6705 | 0.9153 | 0.2029 | 0.6234 | 956 |

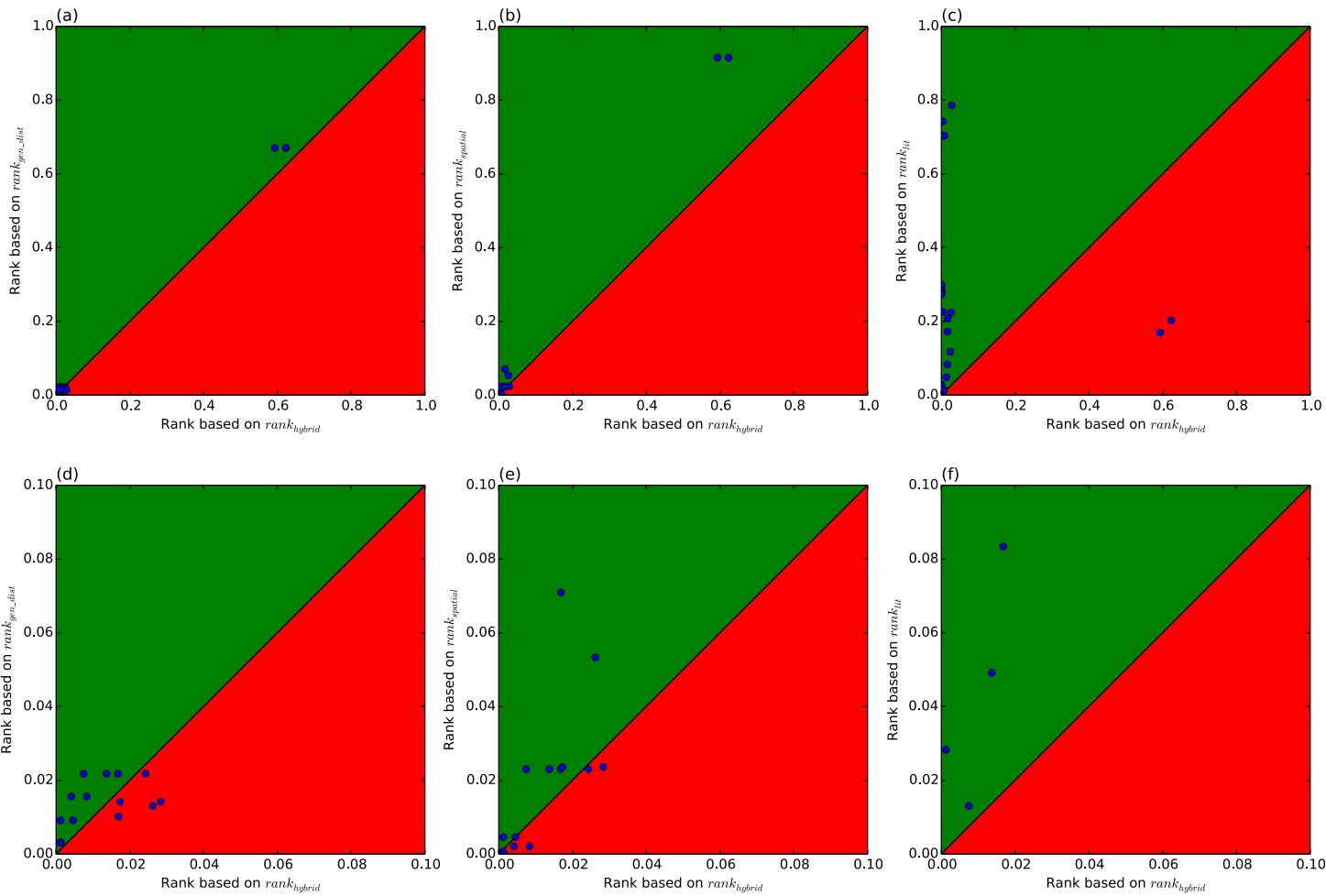

**Figure 1** **Ranks of individual data sources vs. hybrid rank.** For each eQTL SNP, the rank of the target gene based on each of the three sources: genomic distance ($rank_{gen\_dist}$), HiC ($rank_{spatial}$) and literature ($rank_{lit}$), are plotted against the rank of that target gene based on $rank_{hybrid}$. Points in the upper left corner (the green region) occur when the hybrid rank is superior. The hybrid rank is better 12/17, 10/17, 15/17 times as compared to the respective sources. (A)–(C) shows the whole range of ranks [0, 1], whereas (D)–(F) shows the same data zoomed in to the range [0, 0.1].

advantage in 70% of the test cases. Figure 2 further demonstrates that the three sources we have considered are complementary. A pair of correlated sources would result in a scatter plot along the diagonal of the figure.

Since the evaluation set is small, it is difficult to ascertain the statistical advantage of $rank_{hybrid}$. Work is currently underway to obtain a larger eQTL database for validation, but the current results already show the promise of our proposed approach.

We have not included the SNP-gene co-mention strategy ("SNP-gene co-occurrence score ranking") directly in the ranking comparison, as we found it to be too sparse and noisy to be of direct use in ranking. We identified 133 co-mentions of SNPs and genes for the eQTL SNPs, from which 9 match an entry in the eQTL experiments. Since there are 35 unique {SNP, gene} pairs in the eQTL benchmark, we can say that the recall of these is over 25%, even though the precision would be rather low (7%). If we consider only

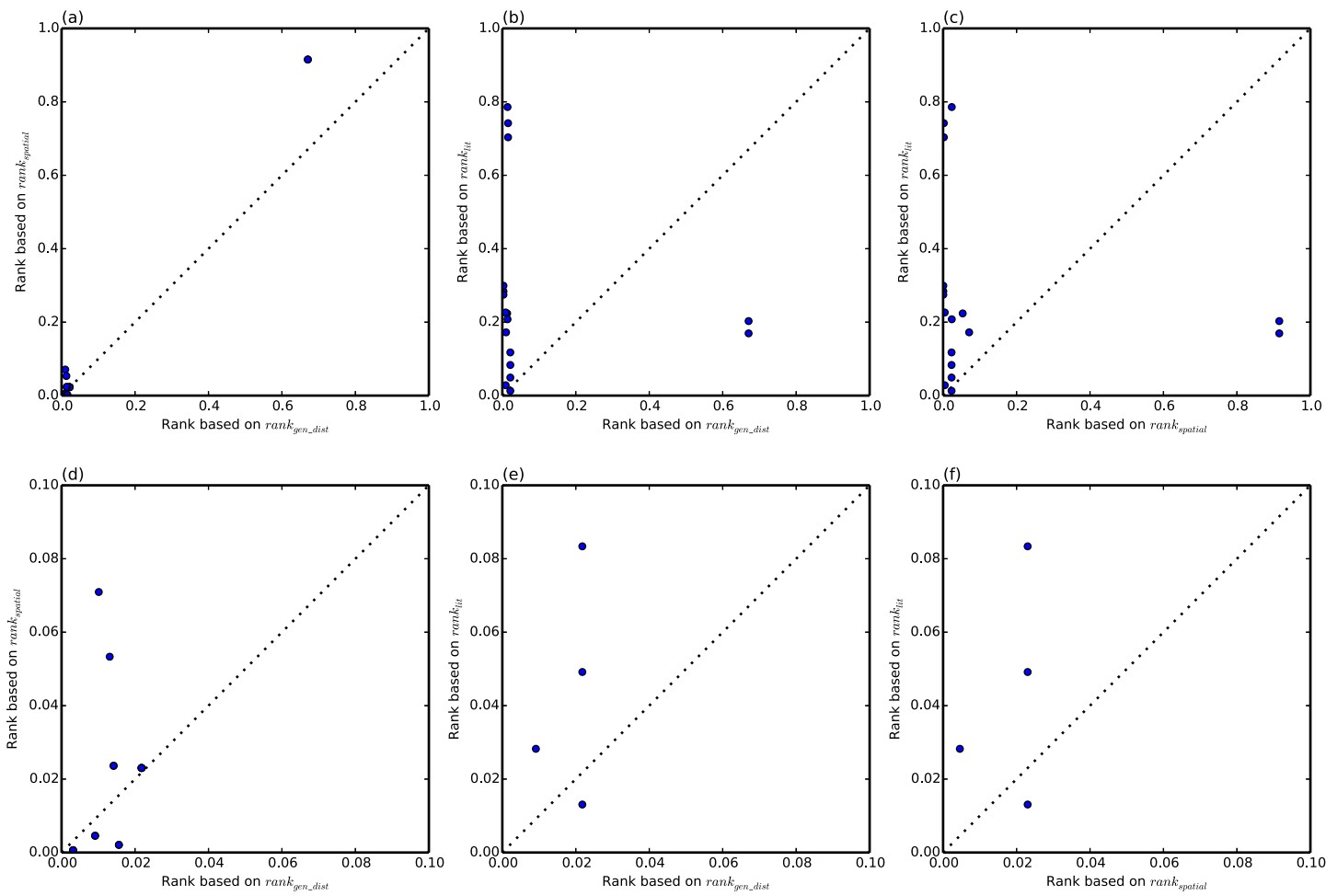

**Figure 2 Pairwise rank comparison of each of the three source rankings.** For each eQTL SNP, each of the three sources of information: genomic distance ($rank_{gen\_dist}$), HiC ($rank_{spatial}$) and literature ($rank_{lit}$), is plotted against each other. (A)–(C) shows the whole range of ranks [0, 1], whereas (D)–(F) shows the same data zoomed in to the range [0, 0.1].

co-mentions with frequency $>1$, there are 3 that match an entry in the eQTL experiments out of 25 co-mentions in total (recall $= 8\%$, precision $= 12\%$). An informal analysis of the results suggests that many genes and SNPs are mentioned together in an abstract that are not biologically or linguistically connected. The precision could be substantially improved by detecting true SNP-gene relationships expressed in the abstracts rather than simple co-mentions; the strategy used to learn and detect protein-residue relationships could perhaps be applied here (*Ravikumar et al., 2012*).

We note that identified SNP-gene co-mentions that do not match an eQTL experiment may in fact be correct; some co-mentions which are false positives from the perspective of the eQTL data may be valid relationships. For this reason, it is difficult to have confidence in any precision analysis and we have preferred the ranking evaluation presented above.

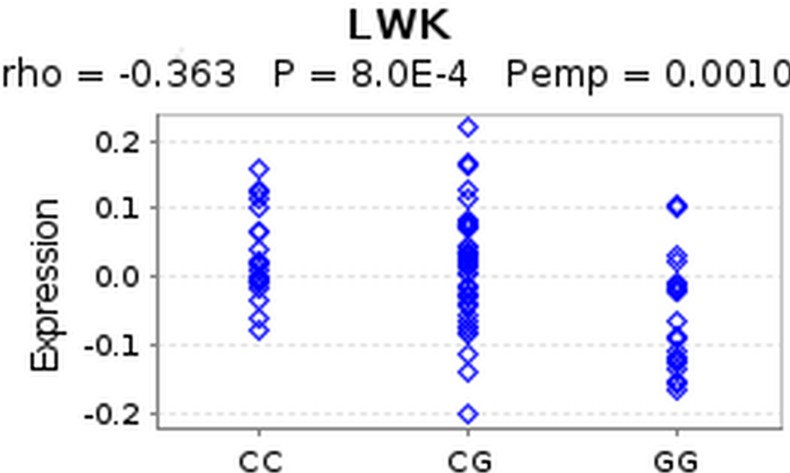

**Figure 3 eQTL of SNP rs4796793 against CEACAM5.** This figure shows an eQTL evaluation of SNP rs4796793 against CEACAM5 for the LWK population using GeneVar.

## Using the hybrid approach for novel SNP target gene discovery

We performed candidate target gene ranking for all of the 67 SNP-disease pairs in our analysis yielding a total of 1,648,736 predicted candidates. We then used our hybrid ranking strategy to prioritise target genes for further exploration. Given that our approach is not limited to selection of target candidate genes on the same chromosome, we filtered our results to look exclusively for candidates on a different chromosome to the SNP, to highlight the power of our method. The top ranked associations can be found in Table 3. We considered SNP/target gene pairs for the top 10 ranked results showing a HiC log ratio greater than 1, combining the ranking from all diseases into one list. The most promising candidate is shown at the bottom of Table 3, linking a risk SNP associated with non-small cell lung cancer (NSCLC) on chromosome 17 with a target gene on chromosome 19, CEACAM5. CEACAM5 has been shown to be over-expressed in lung cancer and was the 6th ranked gene linked to Lung Neoplasms using our literature ranking approach. These two regions from chromosome 19 and chromosome 17 show evidence of interacting from HiC data with a log ratio = 1.47. To determine if this SNP and gene have a regulatory relationship, we searched for supporting evidence from a range of sources.

Firstly, we queried a large-scale eQTL study of HapMap3 (*Stranger et al., 2012*) data via the GeneVar database.[6] In the original study, STAT3 was the reported target gene of SNP rs4796793 (2 kb upstream of the SNP) (*Jiang et al., 2011*). The study showed decreased risk of NSCLC for the minor allele (G). However, when we queried the HapMap eQTL data we found no significant association between STAT3 and rs4796793 across HapMap populations after multiple testing correction. Interestingly, when we examined CEACAM5, we found a significant association with the SNP ($p = 8.0e - 04$) for the Luhya in Webuye (LWK) population in Kenya. The C allele in this case showed increased CEACAM5 expression whereas the G allele showed decreased expression (Fig. 3). This suggests that the C allele, showing increased risk for NSCLC may have a role in increasing expression

[6] http://www.sanger.ac.uk/resources/software/genevar

**Table 3 Data used for SNP target gene discovery.** HiC means the HiC score and Lit refers to the score resulting from literature. rD, rHiC and rL refer to the ranks of genomic distance, HiC and literature respectively. rHy is the geometric mean $rank_{hybrid}$. We report the significant SNPs, confirmed by an independent eQTL study on GeneVar with $p$-value 0.05 of Spearman $\rho$.

| PubMedID | SNP | Disease | chr | Location | Gene | GeneWindow | HiC | Lit | rHiC | rL | rHy | Total | $p$-value |
|---|---|---|---|---|---|---|---|---|---|---|---|---|---|
| 21948749 | rs4796793 | Carcinoma, NSC Lung | chr17 | 37795735 | ERCC1 | chr19:50M-51M | 1.52 | 13.52 | 0.047955 | 0.000258 | 0.000043 | 23230 | 0.0152 |
| 17602083 | rs4796793 | Neoplasm Metastasis | chr17 | 37795735 | CEACAM7 | chr19:46M-47M | 1.47 | 15.74 | 0.062118 | 0.000086 | 0.000043 | 23230 | 0.0459 |
| 23059779 | rs12983047 | Carcinoma, NSC Lung | chr19 | 46526338 | TP53 | chr17:7M-8M | 1.37 | 19.21 | 0.045003 | 0.000174 | 0.000044 | 22976 | 0.0173 |
| 23059779 | rs12983047 | Carcinoma, Squamous Cell | chr19 | 46526338 | TP53 | chr17:7M-8M | 1.37 | 27.91 | 0.045003 | 0.000087 | 0.000044 | 22976 | 0.0173 |
| 21995493 | rs652625 | Lung Neoplasms | chr1 | 12147937 | TTF1 | chr9:134M-135M | 1.49 | 16.41 | 0.004690 | 0.000451 | 0.000045 | 22175 | 0.0231 |
| 22344756 | rs3213182 | Carcinoma, Squamous Cell | chr20 | 31726893 | PDXP | chr22:36M-37M | 1.23 | 19.24 | 0.036609 | 0.000292 | 0.000083 | 23956 | 0.0326 |
| 17602083 | rs4796793 | Carcinoma, Renal Cell | chr17 | 37795735 | XRCC1 | chr19:48M-49M | 1.53 | 73.29 | 0.046965 | 0.000043 | 0.000086 | 23230 | 0.0149 |
| 21948749 | rs4796793 | Lung Neoplasms | chr17 | 37795735 | CEACAM7 | chr19:46M-47M | 1.47 | 18.66 | 0.062118 | 0.000172 | 0.000086 | 23230 | 0.0459 |
| 21995493 | rs652625 | Carcinoma, NSC Lung | chr1 | 12147937 | ERCC1 | chr19:50M-51M | 1.05 | 13.52 | 0.053890 | 0.000316 | 0.000090 | 22175 | 0.038 |
| 21995493 | rs652625 | Lung Neoplasms | chr1 | 12147937 | CEACAM7 | chr19:46M-47M | 1.10 | 18.66 | 0.041082 | 0.000180 | 0.000090 | 22175 | 0.0359 |
| 21948749 | rs4796793 | Lung Neoplasms | chr17 | 37795735 | CEACAM5 | chr19:46M-47M | 1.47 | 18.66 | 0.061601 | 0.000258 | 0.000129 | 23230 | 0.0008 |

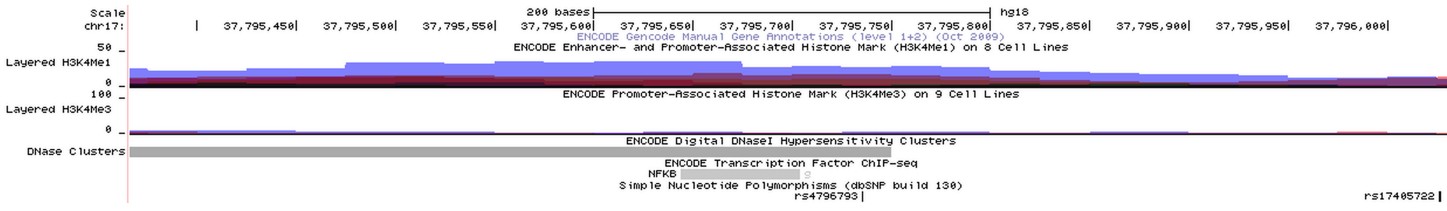

**Figure 4  SNP rs4796793 in the UCSC genome browser.** This figure shows SNP rs4796793 mapped to the UCSC genome browser along with tracks showing ENCODE information for histone modifications, DNaseI Hypersensitivity and transcription factor binding.

of CEACAM5, which is in concordance with the observed increase in expresion in lung cancers (*Blumenthal et al., 2007*). As the HiC data reports these regions interacting in three dimensional space for the major allele, we hypothesise that the minor allele must disrupt this 3D conformation causing decreased expression of CEACAM5 and hence decreased risk. We examined the genomic region containing the SNP and observed and enrichment of histone marks supporting the presence of a regulatory element and in addition observed evidence for an experimentally determined binding site for the transcription factor NFKB (Fig. 4). If NFKB were to bind differentially to either allele, this may explain the change in expression observed in CEACAM5 due to change in the 3D conformation of the DNA. To test if differential binding was likely, we used a computational method that predicts if the allele significantly alters binding affinity of the transcription factor (*Macintyre et al., 2010*). From this the G allele is predicted to have a binding affinity significantly ($p < 5.4e - 06$) stronger than the C allele. This provides a putative functional mechanism for linking a disease-associated SNP with a target gene on a different chromosome. Further experimental work is required to verify this observation.

## DISCUSSION

### Integrating multiple sources of information

One major challenge when integrating diverse sources of information is the various intuitive ways of measuring association. In this work we consider 3D spatial association as a contact probability, 2D spatial association as genomic distance, and literature association as negative log *p*-values. These diverse measures of association can only be numerically combined with appropriate normalisation and transformation, which may be complex and difficult to properly estimate. For example, genomic distance and contact probabilities act in opposite directions, i.e., small distances are "good" but big probabilities are "good". Furthermore, they have different scales as genomic distances could be large integers whereas contact probabilities are in the interval [0, 1]. Using a ranking approach allows different association measures to be converted into the same direction and scale.

The spatial and literature scores complement each other as can be seen in the validation experiment. HiC experiments result in contact map probabilities and hence do not take disease information into account. Literature methods can take disease association into account, but do not include spatial information. For example, the SNP rs344781 has no literature evidence, hence resulting an extremely poor rank for literature. However there is strong HiC evidence for the association (top rank) which results in a good hybrid rank.

**Peer**J

When there is little HiC evidence (such as in rs7096206) weak evidence from literature can pull up the hybrid rank.

The importance of the literature score is further confirmed by Table 2, which shows its value in the discovery setting. Genomic distance cannot be used across chromosomes, and literature evidence provides useful directions, as we have confirmed using GeneVar.

The general approach of integrating various sources of information using the geometric mean is not limited to two sources of information. As different facets of the putative SNP-gene association or SNP-disease association can be estimated, these can all be combined using the geometric mean to obtain a single candidate list of associations for further study.

## Suitability of the eQTL reference set for evaluation

We opted to test the performance of our SNP target gene ranking method using eQTL information as a gold standard. While we considered this the best independent information currently available associating SNPs and genes on a genomic scale, there are some pitfalls that need to be made clear. Firstly, many of the eQTL studies have an inherent bias towards associations that are proximal to the SNP. Due to multiple testing correction burden, it is not possible to test all SNPs against all genes for association, therefore only genes nearby the SNP are tested. In this case, we may be artificially penalising our method against selecting interactions at a large distance. That is, not matching a prediction in the eQTL experiments does not imply that the SNP-gene association predicted is necessarily incorrect, favouring the use of a ranking evaluation instead of precision/recall measures. In addition to this, significant eQTL assocations can arise from a SNP directly affecting a gene, or indirectly affecting the gene via an upstream target. Therefore, these data may also contain false positives with respect to predicting direct targets. Nevertheless, until large scale functional chromatin interaction experimental data is available, we believe eQTL data provides an adequate source of benchmarking data for methods such as those presented in this paper.

## Limitations of the literature-based methods
### Recognition of intergenic SNPs

To focus specifically on intergenic SNPs, it is necessary to categorise SNPs as intergenic or not. The Human Genome Variation Society (HGVS) nomenclature (*Den Dunnen et al., 2000*) provides specific notation for non-coding RNA reference sequences (with the prefix "n."), as well as for referencing specific nucleotides in any genomic reference sequence (with the prefix "g."), where the locations may or may not be coding DNA. This nomenclature in principle provides a technique for straightforward identification of intergenic regions, by looking for genomic variants and cross-referencing their positions with the gene starts/stops in the reference sequences. However, our prior experience with text mining of genetic variants (*Jimeno Yepes & Verspoor, 2014b*) suggests that these descriptors are extremely rarely used in practice. The vast majority of references to genetic variants in the literature refer to coding DNA or protein variants. Our approach to SNP identification, therefore, is limited to recognising known intergenic SNPs ("Intergenic SNP

mention detection"). Other methods could be explored to extend the set of SNPs extracted from text (*Krallinger et al., 2009*; *Thomas et al., 2011*).

### Reliance on named entity recognition for literature identification

As indicated above, there is prior work on using the biomedical literature to support SNP interpretation (*Raychaudhuri et al., 2009*; *Johansson et al., 2012*). Both of these methods rely on links from the NCBI Gene database to identify relevant literature; in contrast we do not rely on the existence of such curated links and perform indexing of direct gene mentions throughout the available abstracts. We are not aware of a study that has specifically considered the completeness of the links to the literature available from the NCBI from the Gene or GeneRIF (Gene Reference Into Function) resources, but given that they must be manually updated or submitted by a member of the community, it is likely that the citations linked there represent only a fraction of the literature relevant to a given gene. A brief investigation of these resources for a single well-studied gene, Human BRCA1, indicates that there are 1,956 PubMed citations associated to its NCBI Gene record. In contrast, our annotation of the abstract text identifies well over 7,000 abstracts that reference that gene. In contrast, for less well-studied genes, we found that there could be links to the literature in the NCBI Gene record that we did not identify from abstract mentions, likely due to mentions of the gene in the full text of the article rather than the abstract, or due to missing name variants/synonyms in our gene name dictionary.

### Alternative methods for establishing gene-disease associations from the literature

Various strategies for establishing gene-disease associations have been explored previously; while we have tested one approach, it could be straightforwardly substituted with an alternative knowledge-based method for establishing gene-disease associations. There are literature-based methods that have addressed association of genes to disease based on co-occurrences for a set of articles defined by a user query (*Rebholz-Schuhmann et al., 2007*; *Tsuruoka, Tsujii & Ananiadou, 2008*) or that rely on matching of MeSH profiles (*Cheung et al., 2012*; *Xiang et al., 2013*). Other methods use indirect relations from co-occurrence analysis (*Frijters et al., 2010*) or based on interaction networks (*Gonzalez et al., 2007*; *Özgür et al., 2008*). Some combine different sources of information including the literature, such as Endeavour (*Aerts et al., 2006*), DisGeNET (*Bauer-Mehren et al., 2010*), G2D (*Perez-Iratxeta et al., 2005*), MimMiner (*van Driel et al., 2006*), PolySearch (*Cheng et al., 2008*) or the approach proposed by *Tiffin et al. (2005)*. Many of these methods could be effectively integrated into our approach.

An alternative to utilising MeSH annotations to establish the association to diseases in the literature would be to do direct annotation of disease mentions in the text, as we have done for gene names. This is the strategy employed by *Johansson et al. (2012)* and some of the other approaches introduced above. However, any automated name recognition strategy will suffer from some rate of noise. For the purposes of experimenting with the literature-based concept, we felt that it was sufficient to utilise the MeSH annotations provided in MEDLINE. Furthermore, we verified that the key terms available from the citation meta-data matched the disease linked with the SNPs in our data. Text-level

annotation of disease names would be appropriate for a method that requires explicit mentions of diseases, e.g., sentence level co-occurrence or information extraction methods.

### *Potential bias stemming from the literature*

As discussed by *Raychaudhuri et al. (2009)*, any literature-based method is dependent on the completeness of the literature and indeed the literature ultimately reflects only those experiments that have been performed. Particularly for the SNP-gene co-mention literature method that we propose, sparseness of mentions in the literature is a problem. This is compounded by the fact that many mentions of genetic variants appear only in the supplementary materials of publications (*Jimeno Yepes & Verspoor, 2014a*) and are therefore difficult to extract using standard text mining strategies. For these reasons, the co-mention strategy is not employed in our evaluation.

As an additional comment, due to the reliance in prior work on genomic distance or eQTLs to establish SNP-gene associations, we would expect that the literature would show bias favoring such associations. This is an additional reason to prefer the use of gene-disease associations in this context rather than direct mentions of SNP-gene associations. Utilising these indicative associations provides indirect evidence; as we have seen this evidence appears to provide a broader context for establishing associations in a disease context.

## CONCLUSIONS

The functional interpretation of intergenic genetic variants poses a challenge to current methods for studying the genetic basis of disease, despite their prevalence in the genomes of individuals. Of particular difficulty are intergenic SNPs that do not appear to have a relationship with the closest gene but instead impact more distant genes, either on the same or even a different chromosome. Such variants cannot be adequately handled by the standard approach of a genomic distance-based ranking. In this work we have explored several strategies for ranking genes as associated to a given SNP; one based on published spatial relationships, another based on general gene-disease relationships mined from the published literature, and proposed a novel third method, in which the two are combined. The focus of our method on ranking genes with respect to genetic variants in intergenic regions is novel with respect to prior work.

We found in a validation study that this hybrid method outperformed a genomic distance baseline in 70% of our test cases, identifying a stronger association between a given SNP and the gene identified in an eQTL study as related to that SNP. This was observed despite an overall bias towards nearby genes in the reference data set. We also considered a direct literature co-occurrence strategy for relating an SNP to a gene, finding that its sparsity made it inappropriate for ranking but that with some refinement it may prove helpful for hypothesis validation. A more exploratory analysis specifically of putative interchromosomal relationships indicates that the hybrid method has substantial promise to support hypothesis generation, identifying potential relationships of SNPs to genes that should be prioritised for experimental validation.
Our validation study is small, due to limited availability of appropriate test data, and our discovery experiment is merely suggestive. There remain many questions about the potential relationship of the genes that our ranking methods identify as strongly associated to the SNPs we have studied, for which no eQTL data is available. These may very well be important new SNP-gene relationships worthy of targeted study.

We have observed the potential of our proposed methods specifically in assisting the interpretation of SNPs interacting with genes on different chromosomes. These are cases that are particularly difficult for any methods that assume proximity of SNPs to the genes they impact to characterise. In future work, in addition to refining the methods with respect to substantially more data and furthering our understanding of their performance, and exploring alternative methods for the constituent parts of the hybrid method in more depth, we plan to build on this observation to develop an architecture that enables characterisation of the full spectrum of intergenic SNPs, by determining criteria that will allow automatic selection of the appropriate ranking method in particular circumstances. That is, rather than insisting that there is a one-method-fits-all-variants ranking solution, we aim to handle more cases more effectively by acknowledging and modelling the full complexity of the problem space.

### Funding
We have received funding from National ICT Australia (NICTA) and the University of Melbourne, Australia. NICTA is funded by the Australian Government through the Department of Communications and the Australian Research Council through the ICT Centre of Excellence Program. NICTA is also funded and supported by the Australian Capital Territory, the New South Wales, Queensland and Victorian Governments, the Australian National University, the University of New South Wales, the University of Melbourne, the University of Queensland, the University of Sydney, Griffith University, Queensland University of Technology, Monash University and other university partners. The funders had no role in study design, data collection and analysis, decision to publish, or preparation of the manuscript.

### Grant Disclosures
The following grant information was disclosed by the authors:
National ICT Australia (NICTA).
University of Melbourne, Australia.

### Competing Interests
The authors declare there are no competing interests.

### Author Contributions
- Geoff Macintyre and Antonio Jimeno Yepes conceived and designed the experiments, performed the experiments, analyzed the data, contributed reagents/materials/analysis tools, wrote the paper, prepared figures and/or tables, reviewed drafts of the paper.

- Cheng Soon Ong conceived and designed the experiments, analyzed the data, contributed reagents/materials/analysis tools, wrote the paper, prepared figures and/or tables, reviewed drafts of the paper.
- Karin Verspoor conceived and designed the experiments, analyzed the data, wrote the paper, reviewed drafts of the paper.

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
