# Peer review of "Associating disease-related genetic variants in intergenic regions to the genes they impact"

_PeerJ, doi:10.7717/peerj.639_

## Round 0.1 · original submission · Major Revisions

· Academic Editor

Major Revisions

Your manuscript has been reviewed by three experts in the field and their comments are shown below. Although all of them admit the basic value of this work, they also give substantial criticism. Especially, one of them recommends its rejection because of his/her questions on the design of the work itself. Please read carefully the following comments and resubmit the revised manuscript, if you think you can answer their questions reasonably.

·

Basic reporting

The authors' study target is to improve priors of genes on which intergenic SNVs have allele-specific effect by incorporating literature-database based score. The idea is interesting and potentially promising. However this reviewer have some major concerns on the study design and interpretation as below.

Experimental design

The hidden relations between intergenic SNVs and target genes are scored with multiple methods including 3D information, literature-based score and also the physical distance, which is most conventional measure and e-QTL-based measure which is adopted as "gold-standard". It might be reasonable to anticipate e-QTL would be most precise but it is still away from "gold-standard", it is better to compare concordance among multiple methods first.
That might be change the word "orthogonal" that the authors used to describe the relation between 3D-method and literature method in the manuscript.
Also the literature-based method is based on previous reports on intergenic SNVs and it means the simple distance would most heavily affect it and e-QTR would be next. It is important to carefully document the baseline relation.

Each method's performance would be better evaluated first and then their combination should be evaluated and why particular combination is chosen, rather than ad hoc should be reasonably described based on the evaluation of individual performance and relations among them.

Selecting "gold-standard" from e-QTL result with simple cut-off of FDR would be losing a lot of information.

Validity of the findings

Multiple testings are considered for the interpretation of results. Words "confirmed" would be too strong. Conclusion would be "The literature-based score and combining it with other ways may be somehow useful to screen target genes or rank target genes with limited usefulness, which might be useful as far as the uses are aware of the limits and benefits".

Additional comments

Nothing to be added.

Reviewer 2 ·

Basic reporting

This paper compared methods to find important SNPs in intergenic regions. Since few researches have systematically studied the functions of the intergenic SNPs, this paper would become a good pioneering work how to prioritize the SNPs for further experiments.

Minor comments:
- Equation (2) for hybrid ranking is a little confusing.
\sqrt{rank_{spatial} \cdot rank_{literature} } is an average score, not rank. You need to convert the score into rank.
- It is not so clear why you need to introduce hybrid ranking between the spatial distance and the literature ranking. If you use the hybrid ranking, why don't you use 2D-literature ranking? Discussion of the correlations between 2D, HiC and literature rankings might be helpful to understand it.
- Please describe SNP IDs of the points in Figure 1, especially for SNPs mentioned in the discussion.
- Calculated original values of 2D, HiC and literature are good resource for biologist and medical scientists. I recommend to provide the values in this paper as a supplemental material or through your web page.

Experimental design

No comments

Validity of the findings

It is difficult to validate the ranking without biological experiment. However, authors checked the consistency of the results with biological knowledge or literatures.

Additional comments

No comments

Reviewer 3 ·

Basic reporting

The authors present an approaching using both the spatial information and literature search to discover disease association for intergenic SNPs. However, there are several important issues that has to be improved before it can be considered for publication.

In the paper, the authors in many instances have cited that they are following the analysis steps of some prior paper. However, in these cases it is still important for the authors to describe the essential details of the cited analysis steps. For example, in Section 2.3.3, the authors state that they used the procedure in Jimeno-Yepes et al., 2013, but failed to provide some detail or discussion on how it is applied to their data.

Futhermore, many other details about their implementation are missing. For example, again in Section 2.3.3, it is stated that they used a dictionary-based approach to annotate the corpus, but no further detail is given on how it is implemented in their model.

Experimental design

In sections 2.3.4 and 2.3.5, the authors discussed how to search for SNP-Disease and Gene-Disease relationships through literature mining. For Gene-Disease relationship, the authors used a T-test to derive a association score, while not testing for the strength of association for SNP-Disease relationships. The authors should offer some explanation for the differences in approach for these two kinds of associations.

Validity of the findings

There are actually quite a few works on eQTL and spatial relationships of the chromosomes, such as Duggal et al NA Research 42(1) 2014, and literature mining using MeSH, such as Xiang et al BMC Systems Biology 2013. Both of these examples use much more complex statistical models in evaluating their search results. The authors should have compared their proposed method to these more advanced methods instead of comparing to the simplistic 2D genomic distance ranking.

---

## Round 0.2 · accepted · Accept

· Academic Editor

Accept

Your revised manuscript has been reviewed by the three original reviewers. Since all of the reviewers are satisfied with the revision, I am happy to inform you that I will accept this manuscript for publishing in PeerJ.

·

Basic reporting

The authors responded to all the comments by this reviewer. This reviewer felt that the authors were optimistic on the bias of literature-based search and non-reliability when truth is unknown. However the authors added descriptions and discussions on the issues.

Experimental design

Nothing to be added.

Validity of the findings

This point was discussed in Basic Reporting.

Additional comments

Nothing to be added

Reviewer 2 ·

Basic reporting

The paper was revised according to my review comment.

Experimental design

No comments

Validity of the findings

No comments

Reviewer 3 ·

Basic reporting

No Comments

Experimental design

No Comments

Validity of the findings

No Comments